# Menopausal Changes in the Microbiome—A Review Focused on the Genitourinary Microbiome

**DOI:** 10.3390/diagnostics13061193

**Published:** 2023-03-21

**Authors:** Min Gu Park, Seok Cho, Mi Mi Oh

**Affiliations:** 1Department of Urology, Inje University Seoul Paik Hospital, 9 Mareunnae-ro, Jung-gu, Seoul 04551, Republic of Korea; 2Department of Urology, Inje University Ilsan Paik Hospital, 170 Juhwa-ro, Ilsanseo-gu, Goyang-si 10380, Republic of Korea; 3Department of Urology, Korea University Guro Hospital, 148 Gurodong-ro, Guro-gu, Seoul 08308, Republic of Korea

**Keywords:** women, microbiome, menopause, gut, vagina, urinary, estrogen, *Lactobacillus*, dysbiosis, genitourinary syndrome

## Abstract

A balanced interaction between the host and its microbiome is crucial to health. Research regarding the significance of the gut and vaginal microbiomes in female health is substantial. However, less data regarding the urinary microbiome are available. Interactions between the gut, vaginal, and urinary microbiomes are also currently being researched. Hormone-induced dysbiosis after menopause is believed to have effects on physical changes and health consequences. Postmenopausal changes in the gut microbiome are associated with increased short-chain fatty acids and hydrogen sulfide levels. Increased vaginal pH caused by reduced estrogen alters the vaginal microbiome, resulting in reduced levels of *Lactobacillus*. Such changes influence the vaginal structure and functions, contributing to the onset of genitourinary syndrome of menopause. A dysbiosis of the urinary microbiome is associated with urgency and urinary incontinence and also related to interstitial cystitis/bladder pain syndrome and neuropathic bladder. As these diseases commonly affect postmenopausal women, hormone-induced changes in the microbiome may play a role. Menopause increases the alpha diversity of the urinary microbiome and lowers the percentage of *Lactobacillus* in urine, and such changes precede recurrent cystitis. More research regarding the effects of changes in the urinary microbiome due to menopause on urinary tract diseases is needed.

## 1. Introduction

Humans are host organisms with stable and transient symbiotic microbiota [1]. A balanced interaction between the host and microbiota is crucial to maintaining host health [2]. Several studies have investigated the effects of the gut microbiota on human health and diseases, leading to studies investigating the microbiomes of the vagina and skin. However, data regarding the urinary tract microbiota are lacking [3].

Menopause is the conclusion of the menstrual cycle in relation to ovarian failure due to reduced estrogen secretion and the loss of progesterone secretion, and with increased life expectancy, the postmenopausal period may account for up to one-third of a woman’s life [1]. The dramatic changes in a woman’s health after menopause are well-known, and menopause alters the gut microbiota, which has several effects on a woman’s health [4]. In the ovulatory cycle, estrogen and progesterone cause thickening of the stratified squamous epithelium of the vagina, deposition of glycogen, and local immunity; therefore, these hormones have substantial effects on the composition of the microbiome [5]. Alterations in the microbiota due to postmenopausal hormone changes are similar to vaginal dysbiosis observed in inflammatory pelvic disease, human immunodeficiency virus and human papillomavirus (HPV) infections, and pregnancy [1].

Further research is required to determine the role of the urinary microbiome in maintaining a structural and functional homeostasis of the urinary tract [6]. Several studies have reported that the urinary microbiome is associated with various diseases of the urinary system, including overactive bladder [7,8], interstitial cystitis/bladder pain syndrome (IC/BPS) [9], bladder cancer [10], and urinary tract infections [6]. These findings suggest that the urinary microbiome plays an important role in urinary tract disease [6].

Urinary pathogens are believed to originate in the gastrointestinal tract, with an intermediary step of vaginal colonization [11,12]. Approximately 62.5% of intestinal-derived species and 32% of vaginal-derived species overlap with species in the urine [13]. These common pathogens support the origination of the urinary microbiome from the gut and vagina, and a mutual, symbiotic relationship of these strains suggests that diseases can occur due to crosstalk between each environment. Therefore, changes in the gut and vaginal microbiome as a result of dramatic hormonal changes during menopause will impact the urinary microbiome. The changes of the urinary microbiome require investigation in relation to the pathophysiology of various urinary tract diseases that commonly affect postmenopausal women.

An understanding of the changes in the microbiome from the perspective of postmenopausal changes and the correlations between the microbiome and postmenopausal changes will help identify new methods for alleviating and controlling various postmenopausal symptoms, including the genitourinary syndrome of menopause (GSM). Moreover, these findings will provide more insight into functional urological diseases. In this review, we discuss postmenopausal changes in the gut, vagina, and urinary microbiomes, the possibility of their interactions, and their effects on postmenopausal health. We particularly focus on postmenopausal changes and the urinary microbiome, which are lesser studied than the gut and vaginal microbiome.

## 2. Changes in Gut Microbiota in Menopausal Women

It has been reported that diversity in intestinal microbiome and functions can affect various estrogen-dependent disorders by altering estrogen concentrations independent of the ovaries [14]. Conversely, several studies have investigated the effects of postmenopausal changes in female hormones on the gut microbiome. For instance, a study by Auriemma et al. examined changes in the vaginal and gut microbiome in 35 pre- and postmenopausal women and reported that postmenopausal women had significantly lower vaginal *Lactobacillus* compared to premenopausal women, but their overall diversity in gut flora did not significantly differ from that of premenopausal women [15]. Furthermore, another study compared the microbiomes of the small intestine in postmenopausal women receiving hormone therapy (n = 13), postmenopausal women not receiving hormone therapy (n = 12), and reproductive-aged women (n = 10) [16]. Leite et al. found that the core duodenal microbiomes differed between those who received hormone therapy and those who did not [16]. Moreover, those receiving hormone therapy had similar microbiome compositions to those of the premenopausal group. Those not receiving hormone therapy had a higher abundance of Proteobacteria but lower *Bacteroidetes* and duodenal microbial diversity compared to those receiving hormone therapy. Additionally, the hormone therapy group had relatively higher Prevotella and lower *Escherichia* and *Klebsiella* [16]. Thus, in contrast to the study by Auriemma et al. [15], these results suggest that menopause-induced hormonal changes significantly alter the gut microbiome and that hormone therapy can restore the microbiome to some degree. Santos-Marcosa et al. [17] also reported that postmenopausal women have a higher *Firmicutes/Bacteroidetes* ratio and a higher relative abundance of *Lachnospira* and *Roseburia* compared to premenopausal women, while premenopausal women have a lower relative abundance of the *Prevotella*, *Parabacteroides*, and *Bilophila* genera. Yang et al. [4] performed a meta-analysis to examine changes in the intestinal microbiota in menopausal women. While no significant changes were observed at the phylum level, such as changes in *Firmicutes*, *Bacteroidetes*, or *Proteobacteria* compositions, genera *Odoribacter* and *Bilophila* were more abundant in menopausal women than in premenopausal women. The prevalence of *Sutterella*, *Roseburia*, and *Blautia* was decreased, while *Prevotella*, *Parabacteroides*, and *Bacteroidetes* were more common in menopausal women, though these differences were not significant. Hydrogen sulfide produced by *Biophila* spp. relaxes the ileal smooth muscles and increases the blood supply of the gastrointestinal mucosa [18]. Increased *Bilophila* in menopausal women leads to increased hydrogen sulfide production, inducing local inflammation and mucosal damage, increased serum endotoxin concentrations, and inflammatory reactions in several types of tissues. The intracellular inflammatory response caused by inflammatory factors leads to insulin resistance by interfering with insulin signal transduction. In bones, inflammatory factors (such as tumor necrosis factor-α, interleukin (IL)-1, and IL-6) enhance the function of osteoclasts, contributing to osteopenia. Inflammatory factors in peripheral blood cross the blood–brain barrier and activate microglia in the central nervous system, resulting in inflammation in neurons and exacerbating the aggregation and accumulation of nerve fiber tangles and β-amyloid protein, which contribute to Alzheimer’s disease [4].

Increased *Odoribacter* in menopausal women results in elevated levels of short-chain fatty acids (SCFAs), hydrogen, and hydrogen sulfide [19,20]. SCFAs increase fatty acid oxidation and energy metabolism, are involved in the synthesis of serotonin and stabilizing neurons, and increase circulating insulin-like growth factor-1, which stimulates osteogenesis. Thus, *Odoribacter*-related increases of SCFAs in menopausal women may lower the risk of obesity, hyperlipidemia, depression, and osteoporosis. In contrast, increased hydrogen sulfide production leads to inflammatory reactions. Therefore, *Odoribacter* has both positive and adverse effects (similar to the effects of postmenopausal syndrome) [4].

Furthermore, a study that analyzed the gut microbiome of menopausal women with and without menopausal syndrome at time of menopause found that those with menopausal syndrome exhibited gut microbiome imbalance due to a deficiency of *Aggregatibacter segnis*, *Bifidobacterium animalis*, and *Acinetobacter guillouiae* [21]. Regarding osteoporosis, which commonly affects menopausal women, Yang et al. [22] reported significant differences in the gut microbiome associated with the severity of osteoporosis. Those without osteoporosis had an abundance of *Romboutsia*, unclassified_*Mollicutes*, and *Weissella* spp. In contrast, those with osteoporosis had an abundance of *Fusicatenibacter*, *Lachnoclostridium*, and *Megamonas* spp. Changes in the gut microbiome were more pronounced than those of the vaginal microbiome in the various severities of osteoporosis, suggesting that the gut microbiome of women may influence bone metabolism [22].

## 3. Changes in Vaginal Microbiota in Menopausal Women

The composition of vaginal microbiota is highly dynamic, and it is affected by age, ethnicity, physiological factors, and the immune system. Vaginal infections, medications, as well as lifestyle and diet affect the vaginal microbiota [23]. The vaginal microbiome changes throughout a woman’s lifetime, and such changes have an important impact on quality of life [24]. Before describing the postmenopausal vaginal microbiome, we summarize the changes in the vaginal microbiome from birth, throughout puberty, and in reproductive periods (Table 1) [23,24].

The vaginal microbiome composition is altered after menopause, although there are individual variations. Generally, the vaginal microbiota includes *Gardnerella vaginalis*, *Ureaplasma urealyticum*, *Candida albicans*, and *Prevotella* spp., while the prevalence of *Lactobacillus* progressively declines [25,26]. The reduction of lactic acid bacteria in menopausal women is a normal physiological change [27], and changes in the vaginal microbiome in menopausal women can impair sexual health by causing vulvovaginal atrophy and vaginal dryness [27].

According to a study that analyzed the differences in the vaginal microbiome in Korean pre- and postmenopausal women [28], postmenopausal women had significantly fewer Lactobacillus and significantly more *Prevotella*, unclassified *Lactobacillaceae*, *Escherichia*, *Pseudomonas*, *Proteus*, *Finegoldia*, and *Atopobium*. Hillier et al. [29] examined the vaginal microflora of 73 menopausal women who had not undergone hormone replacement therapy, and these women had markedly lower percentages of *Lactobacillus* and vaginosis-associated bacteria compared to women of childbearing age. The prevalence of bacteria were 49% *Lactobacilli*, 27% *Gardnerella vaginalis*, 13% *Ureaplasma urealyticum*, 1% *Candida albicans*, and 33% *Prevotella bivia* [30]. Cauci et al. [31] reported that the prevalence of bacterial vaginosis is lower (6%) among menopausal women than among women of childbearing age (9.8%) or perimenopausal (11%) women. Mirmonsef et al. [32] analyzed the differences in the types of vaginal glycogen and *Lactobacillus* in the cervicovaginal lavage samples of 11 premenopausal women and 12 postmenopausal women. *L. jensenii* levels were significantly correlated with free glycogen levels in all samples. *L. iners* levels were significantly higher in premenopausal women than in menopausal women. According to Lee et al. [33], menopausal women had a significantly lower free glycogen concentration than premenopausal women, and premenopausal women had higher *Lactobacillus* levels and lower vaginal pH than menopausal women. Overall, a positive correlation between *L. iners* and glycogen levels was identified in both premenopausal and postmenopausal women. In a cross-sectional observational study to identify the main characteristics of the vaginal microbiota in healthy postmenopausal women and to examine the influence of rectal lactobacilli, it was confirmed that rectal lactobacilli maintain a normal vaginal microbiota by playing the role of a reservoir in menopausal women lacking vaginal lactobacilli [34].

Emerging evidence shows that the uterus harbors its own microbiome, which can modify uterine functions and disease [35]. There is also rapidly accumulating evidence for the role of the postmenopausal vaginal microbiome in endometrial cancer. In a study of the microbiome in cervicovaginal and anorectal swab samples from menopausal women with endometrioid adenocarcinoma, uterine serous carcinoma, and benign uterine conditions, the microbial diversity of anatomical ecological niches with endometrioid adenocarcinoma and uterine serous carcinoma was found to be different compared with that in benign controls [36]. Walsh et al. [37] identified the postmenopausal status as the main driver of a polymicrobial network associated with endometrial cancer microbiome in their clinical study on 151 patients undergoing hysterectomy. The presence of Porphyromas somerae is the most predictive microbial marker of endometrial cancer and might be useful in detecting endometrial cancer in high-risk, asymptomatic women. Wang et al. [38] revealed the significant differences between the paired endometrial cancer and adjacent non-endometrial cancer endometrial microbiota in their clinical study of 28 menopausal women who were histopathologically diagnosed with endometrioid adenocarcinoma, some of whom underwent hysterectomy using the open surgical approach.

Some recent clinical trials administered *Lactobacillus* to postmenopausal women. Ribeiro et al. [39] randomly divided 60 menopausal women into three groups and provided different interventions: isoflavone alone, isoflavone with probiotics (*L. acidophilus*, *L. casei*, *Lactococcus lactis*, *Bifidobacterium bifidum*, and *Bifidobacterium lactis*), and hormone therapy (1 mg estradiol and 0.5 mg norethisterone acetate). The vaginal pH and *Lactobacilli* levels were restored to premenopausal levels in patients who underwent hormone therapy, though no significant changes were observed in the other groups [39]. In a systematic review of four clinical trials investigating the effects of percutaneous estrogen administration on the vaginal microbiome in postmenopausal women, Ratten et al. [40] found that topical estrogen administration increases vaginal *Lactobacillus*. In a randomized, double-blind, placebo-controlled study conducted by Petricevic et al., when probiotic capsules containing L. rhamnosus GR-1 and L. reuteri RC-14 were administered daily for 14 days, there was a significant improvement in the Nugent score compared with that in the placebo group. The results suggest an alternative modality to restore the normal vaginal flora using oral administration of specific lactobacilli strains [41].

## 4. Changes in Urinary Microbiota in Menopausal Women

The clinical significance and functions of the urinary tract microbiome have not been extensively studied. However, some studies have reported that dysbiotic urinary microbiota is associated with urge urinary incontinence (UUI) [7,42,43,44], overactive bladder (OAB) [45], neuropathic bladder [46], IC/BPS [47] and recurrent urinary tract infections (rUTI) [48]. However, the causal relationships between altered urinary microbiota and urinary tract diseases remain elusive. The risks of UUI, OAB, IC/BPS, and rUTI increase with advancing age and after menopause [49,50,51]. The reason for the high incidence of urinary tract disease after menopause is unclear, though altered urinary microbiota as a result of age or hormone changes likely contribute to some degree [52,53,54,55]. As noted, age-related or menopausal/hormone-related changes in the microbiota have been observed in other microbial niches, including those in the gut and vagina [56,57]. However, as shown in Table 2, studies on the urinary microbiome in postmenopausal women have not been extensively conducted.

Ammitzbøll et al. [58] analyzed urinary microbiota composition via 16S rRNA gene sequencing of catheterized urine samples of menopausal women to determine the effects of menopause on the urinary microbiota and found that urine from menopausal women had a substantially higher alpha diversity compared to urine from premenopausal women. *Lactobacillus* was the most abundant bacteria in both groups, although the relative abundance of *Lactobacillus* was 77.8% in premenopausal women and 42.0% in postmenopausal women. Premenopausal women had *Lactobacillus*-dominant urotypes, while postmenopausal women had a more diverse urinary microbiota with more abundant *Gardnerella* and *Prevotella* genera [58].

Hugenholtz et al. [59] compared the urine and vaginal microbiota compositions of pre- and postmenopausal women, those with recurrent UTI (RUTI), and those who had undergone renal transplantation. There was a difference in urine and vaginal microbiota composition between the pre- and postmenopausal groups, where postmenopausal women had a lower abundance of *Lactobacilli* and gram-negative uropathobionts, and a higher abundance of bacterial vaginosis anaerobes and gram-positive uropathobionts. There were no significant differences in the microbiome between those who had RUTI and renal transplantation. Thus, postmenopausal status has a stronger impact on microbiome composition than other factors. Additionally, RUTI was generally caused by *Escherichia/Shigella* in premenopausal women, but the causative pathogen of RUTIs were varied in postmenopausal women.

Curtiss et al. [53] analyzed the bladder microbiome using urine samples from 79 healthy women and identified 60 different genera. Although the number of genera decreased with advancing age, the changes were not statistically significant. The prevalence of *Lactobacillus* was significantly lower in menopausal women than in premenopausal women, and the prevalence of *Mobiluncus* was significantly higher in menopausal women than in premenopausal women.

The clinical significance of postmenopausal changes in the urinary microbiome is suggested by the prevalence of urinary tract diseases. Bossa et al. reported that changes in the urinary microbiome precede the onset of urinary tract infections, and the normal urinary microbiome is restored after urinary tract infections are resolved [60]. Morand et al. [61] reported the taxa percentages of major phyla in the human urinary tract to be *Proteobacteria*, 35.6%; *Firmicutes*, 31.3%; *Actinobacteria*, 22.4%; *Bacteroidetes*, 6.4%; and others, 4.3%. Therefore, it was concluded that the majority of pathogenic bacteria are commensal in the human urinary tract and that their pathogenicity is induced by an imbalance in the relative abundance ratios [61]. Menopause, which has a significant impact on the urinary microbiota environment, may facilitate the pathogenicity of pathogenic bacteria.

Vaughan et al. [62] conducted a cross-sectional study including patients with recurrent cystitis who were administered prophylactic antibiotics, patients with recurrent cystitis who did not undergo antibiotic prophylaxis, and age-matched controls of menopausal women (mean participant age: 55 years). All three groups were undergoing estrogen therapy after menopause. No significant differences were observed between the groups in the number of bacteria in the microbiome, including *Lactobacillus*. This result may be attributable to the fact that all three groups were undergoing hormone therapy after menopause. Significant differences in anaerobic taxa associated with phenotypic groups were identified in the previous study. Most of these differences included *Bacteroidales* and the *Prevotellaceae* family, though differences in *Actinobacteria* and certain genera of *Clostridiales* were also noted. Yoo et al. [63] examined the differences in the microbiomes of patients with acute uncomplicated cystitis (AUC) and patients with recurrent cystitis (mean patient age: 54 years) and found no significant differences in the microbiomes between the groups. The percentage of menopausal women was 54.5% in the AUC group, 67.7% in the recurrent cystitis group, and 64.2% in the total study population. Microbiome diversity was significantly higher among patients with recurrent cystitis than in those with AUC, and the microbiome composition was significantly different between the groups. In the urine next generation sequencing, *Pseudomonas*, *Acinetobacter*, and *Enterobacteriaceae* were identified in the AUC group, and *Sphingomonas*, *Staphylococcus*, *Streptococcus*, and *Rothia* spp. were detected in patients with recurrent cystitis. AUC is a temporary infection caused by a specific causative organism, while dysbiosis seems to play a more important role in the pathophysiology of recurrent cystitis [64]. In addition, there is a possibility that the dysbiosis of patients with recurrent cystitis changes as the condition is prolonged. The increased prevalence of recurrent cystitis among menopausal women may be due to an acceleration of dysbiosis as a result of menopause-related hormone changes.

Differences in urinary microbiome between IC/BPS patients and normal controls have been investigated in several studies. Abernethy et al. [65] suggested that Lactobacillus strains and higher levels of proinflammatory cytokines are associated with IC/BPS [65]. However, in other studies, a midstream urine sample from IC/PBS patients and asymptomatic controls showed no significant difference in the presence, variety, and abundance of species and genera across groups [66,67]. Therefore, further research is necessary to arrive at a consensus. Moreover, there is lack of study on the characteristics of urinary microbiome in postmenopausal women with IC/BPS. Considering the worsening of IC/BPS symptoms with the onset of the menstrual period in many women, low estrogen level is likely to play a role in IC/BPS [68]. Furthermore, there is some evidence that estrogen affects immune cells, such as Th1, Treg, Th17, dendritic, and mast cells [69,70,71]. Therefore, postmenopausal changes in estrogen levels clearly affect the pathophysiology of IC/BPS. More research on the effect of postmenopausal changes in urinary microbiome on IC/BPS is warranted.

Postmenopausal women may undergo local hormone therapy for primary or secondary prevention of urinary tract infections [11,72]. Lillemon et al. [73] investigated the effects of a 12-week estrogen-containing vaginal ring on the bladder microbiome in postmenopausal women and observed no significant changes after treatment. However, other studies found that local vaginal estrogen therapy increased vaginal [74] and bladder [75] *Lactobacillus* counts in menopausal women, and local vaginal estrogen supplementation and *Lactobacillus* supplements decreased the risk of urinary tract infections among patients with recurrent urinary tract infections [74,76,77,78]. Anglim et al. [79] reported that local estrogen therapy alters the local hormonal environment of the urinary bladder and thus reduces *Finegoldia magna* in the bladder, thereby lowering the incidence of RUTI in postmenopausal women.

A systemic review reported that all commercially available vaginal estrogen products are useful for urinary urgency, urinary frequency, nocturia, and recurrent urinary tract infections [80]. Previous studies have reported that vaginal microbiome transplantation from a healthy donor can restore a *Lactobacillus*-dominant vaginal microbiome and prevent urinary tract infections in women with refractory or recurrent bacterial vaginitis [59,81,82].

**Table 2 diagnostics-13-01193-t002:** Summary of studies on the urinary microbiome in postmenopausal women.

Study	Year	Subjects (n)	Specimens	Analysis Technique	Meaningful Results
Ammitzbøll et al. [58]	2021	41 premenopausal and 42 postmenopausal women	Catheterized urine samples	16S rRNA gene sequencing	More diverse urinary microbiota with higher abundance of the genera *Gardnerella* and *Prevotella*
Hugenholtz et al. [59]	2022	18 premenopausal controls and 18 premenopausal recurrent UTI (RUTI) cases, and 30 postmenopausal controls and 20 postmenopausal RUTI cases with and without a renal transplant	Self-collected midstream urine and vaginal flocked swab	16S rRNA gene sequencing	Little relative abundances of lactobacilli (*L. crispatus*), lower gram-negative uropathobionts, and higher bacterial vaginosis anaerobes and gram-positive uropathobionts in urine and vaginal samples
Jung et al. [76]	2022	17 postmenopausal women with RUTI	Clean-catch urine samples	16S rRNA gene sequencing	*Lactobacillus* increases in the urogenital microbiome of postmenopausal women with RUTI after 6 months of vaginal estrogen. Relative increase in *L. crispatus* specifically is associated with treatment success
Curtiss et al. [53]	2018	60 healthy premenopausal and 19 postmenopausal women	Clean-catch mid-stream urine	16S rRNA gene sequencing	*Lactobacillus* is more common in premenopausal women and *Mobiluncus* is more common in postmenopausal women
Anglim et al. [79]	2022	37 postmenopausal women with (n = 17) and without (n = 20) RUTI	Catheterized urine sample at recruitment and 3–6 months following treatment with local estrogen therapy	16S rRNA gene sequencing	*Klebsiella aerogenes* in 80% of RUTI group and in 53.3% of control group, abundance of *Finegoldia magna* was decreased from 33.3% to 6.7% after local estrogen therapy
Lillemon et al. [73]	2022	39 postmenopausal women randomly divided into two groups (placebo vs. estrogen)	Catheterized urine and mid-vaginal swab samples at recruitment and 12 weeks following treatment	16S rRNA gene sequencing	No significant change in the bacterial composition of the vaginal or urinary bladder microbiome
Thomas-White et al. [75]	2020	62 postmenopausal women with overactive bladder	Catheterized urine samples, vaginal and perineal swabs	Expanded Quantitative Urine Culture	Estrogen therapy for overactive bladder resulted in decreased bladder bacterial diversity and increased bladder *Lactobacillus*

## 5. Genitourinary Syndrome of Menopause (GSM)

In a paired sample analysis of 197 vaginal and urinary samples from women (mean participant age: 53 years), Komesu et al. [83] reported that 60 of the 100 most abundant operational taxonomic units in the samples overlapped. Hugenholtz et al. [59] compared the urinary and vaginal microbiomes in premenopausal and postmenopausal women and identified *Lactobacilli*, bacterial vaginosis anaerobes, and gram-positive uropathobionts in urine and the vagina. In addition, *Lactobacilli* dominance was greater in the vagina than in urine, a trend that was more evident among premenopausal women than among postmenopausal women. These results suggest that manipulation of the vaginal microbiome may have a significant impact on the development of lower urinary tract diseases as the vaginal microbiome and bladder or urinary tract microbiomes may directly influence one another [83]. However, studies regarding the interactions between vaginal and urinary microbiomes are lacking [83]. To understand the mutual influence between the vaginal and urinary microbiomes, GSM must be considered. GSM is a collection of symptoms resulting from diminished hormonal stimulation to the vulva, vagina, or lower urinary tract. Approximately 50% of postmenopausal women report symptoms of GSM. These symptoms are typically progressive and unlikely to resolve spontaneously [84]. One study examined the effects of the vaginal microbiome composition on genitourinary menopausal symptoms, serum estrogen, and vaginal glycogen in 88 menopausal women [85] and reported that 66% of women had at least one *Lactobacillus* species and that 38% had a *Lactobacillus*-dominant vaginal microbiome. While 24% had both *L. crispatus* and *L. iners*, 9%, 32%, and 34% had *L. crispatus* only, *L. iners* only, or neither, respectively [85]. GSM was not present in women with *Lactobacillus* species in their microbiome [85]. Furthermore, menopausal women with *Lactobacillus*-dominant microbiomes had higher levels of unconjugated serum estrone, although their vaginal glycogen levels did not significantly differ from those of women with non-*Lactobacillus*-dominant microbiomes [85].

The impact of estrogen on the vaginal microbiome has been the subject of recent research. In the premenopausal vagina, the microbiome is represented by five Community State Types (CST), including type I, with a predominance of *Lactobacillus crispatus*; type II, with a predominance of *Lactobacillus gasseri*; type III, with a predominance of *Lactobacillus iners*; and type V, with a predominance of *Lactobacillus jensenii*. Type IV does not identify *Lactobacillus*, but focuses on a heterogeneous population of bacteria. Type IV-A is composed of a small proportion of *Lactobacillus* and anaerobic bacteria (*Anaerococcus*, *Entomophiles*, and *Prevotella*). Type IV-B has a large proportion of *Atopobium* and also includes *Prevotella*, *Parvimonas*, *Sneathia*, *Gardnerella*, *Mobiluncus*, and *Peptoniphilus* species [86,87]. Studies regarding the microbiome of symptomatic postmenopausal women report that *Lactobacillus* species are diminished, with a shift toward CST IV and increased bacterial diversity [27]. The 20–50% of postmenopausal women with a *Lactobacillus*-dominated vaginal microbiome have a lower prevalence of GSM symptoms. Thus, altering the vaginal microbiome with estrogen or *Lactobacillus* supplementation could be considered as a treatment option for GSM. Lillemon et al. [73] reported that treatment with 12 weeks of vaginal estrogen just changed the main symptoms of GSM from vulvovaginal dryness and urinary frequency to dyspareunia and urgency without significant improvement of GSM symptoms. There was a pilot RCT attempting to improve GSM using the microbiome [88]. The study placed 35 pre- and 35 postmenopausal women into treatment groups that either received *Lactobacillus*-containing feminine soap and cream and/or Lactobacillus-containing feminine gel (4 weeks), or a control group (no treatment). A comparison of the changes in vaginal microbiota and genitourinary symptoms after four weeks showed that those treated with the *Lactobacillus*-containing soap and cream had lower vaginal pH and pathogenic flora compared to the control group. This change was more pronounced in the postmenopausal group. GSM symptoms were improved in 81.3% of postmenopausal women—a significant improvement compared to the control group. The symptom of overactive bladder also improved in 91.7% of postmenopausal women in the treatment group [88]. Jaisamrarn et al. [89] conducted a clinical trial investigating the effects of combined estrogen and *Lactobacillus* therapy. In this study, patients with GSM were divided into the placebo and intervention group that used a vaginal cream containing estriol and *L. acidophilus* (KS400) for 14 weeks. The cream containing estrogen and *Lactobacillus* significantly improved GSM symptoms compared to the placebo and significantly restored vaginal microbiome colonies. Therefore, it is possible that such combination therapy is more effective than estrogen or *Lactobacillus* monotherapy for GSM management (Table 3).

## 6. Conclusions

The gut, vaginal, and urinary microbiomes are mutually influential, and dysbiosis of one of these microbiota results in several health problems. Menopause alters the gut, vaginal, and urinary microbiota due to drastic hormone changes. Therefore, menopause may cause several physical changes and subsequent health consequences. An understanding of the changes in the microbiome and their significance in menopausal women is crucial, and further research is required to analyze their correlations with various diseases. In particular, endometrial cancer appears to be closely related to changes in the postmenopausal vaginal microbiome. In contrast to postmenopausal changes in the gut and vaginal microbiota, research regarding the urinary microbiome is inadequate. Lower urinary tract symptoms or recurrent cystitis that commonly affect postmenopausal women are likely to be associated with changes in the urinary microbiome after menopause. This fact highlights the importance of research regarding menopause-related changes in the urinary microbiome and the clinical implications of these changes. In addition, there is substantial overlap between the vaginal and urinary microbiomes, which directly and indirectly influence one another. Therefore, GSM, a prevalent syndrome among postmenopausal women, must be studied from this perspective.

## Figures and Tables

**Table 1 diagnostics-13-01193-t001:** Changes in the vaginal microbiome during a woman’s lifetime.

	Birth	Childhood	Puberty	Reproductive	Pregnancy
Dominating vaginal microbiome	Wide range of aerobes and facultative anaerobes	Gram-positive anaerobic bacteria: *Actinomyces*, *Bifidobacterium*, *Peptocuccus*, *Peptostreptococcus*, *Propionibacterium* Gram-negative anaerobic bacteria: *Bacteroides*, *Mycoplasma*, *Fusobacterium*, *Veillonella* Gram-negative aerobic bacteria: *Staphylococcus aureus*, *Staphylococcus epidermidis*, *Escherichia coli*, *Streptococcus viridans*, *Enterococcus faecallis*	*L. crispatus*, *L. gasseri*, *L. iners*, and *L. jensenii*, *Atopobium*, *Streptococcus* spp.	*L. crispatus*, *L. gasseri*, *L. jensenii*, *L. iners*, *Anaerobic bacteria*	Increasing lactobacilli dominance, especially *L. iners*, *L. crispatus* (most stable species across pregnancy) Postpartum: *Streptococcus anginosus*, *P. bivia*, *L. iners*. Less diverse and lower abundance of *Lactobacillus* spp.
Vaginal pH	Neutral or alkalized	Neutral or alkaline	Acidic	Lowering of local pH (<4.5)	Acidic

**Table 3 diagnostics-13-01193-t003:** Summary of the studies about therapeutic interventions affecting the postmenopausal vaginal and urinary microbiome.

Study	Year	Subjects (n)	Therapeutic Regimen	Route of Probiotics Administration	Meaningful Results
Ribeiro et al. [39]	2018	60 postmenopausal women	Isoflavone alone, isoflavone with probiotics, estradiol and norethisterone acetate	Oral	Vaginal pH and *Lactobacilli* levels were restored in the hormone therapy group
Shen et al. [90]	2016	Experimental group of 30 postmenopausal women with vaginal atrophy and 29 postmenopausal women without vaginal atrophy as control group	Conjugated low dose estrogen	Oral	Significant increases in the relative abundance of *Lactobacillus* spp. and decrease of *Gardnerella*, two-fold increase in the Vaginal Maturation Index
Dahn et al. [91]	2008	20 postmenopausal women divided into experiment and control groups	Permarin^®^ (conjugated equine estrogen in combination with progesterone)	Oral	Increase of Lactobacillus and significant improvement of Nugent score after estrogen replacement
Petricevic et al. [41]	2008	72 healthy postmenopausal women randomly divided into experimental and placebo groups	Probiotic capsules containing L. rhamnosus GR-1 and L. reuteri RC-14	Oral	Significant improvement in Nugent score compared to the placebo group
Heinemann et al. [92]	2005	40 postmenopausal women divided into experimental and control groups	Permarin^®^ (conjugated equine estrogen in combination with progesterone)	Oral	*Lactobacillus* was more often the dominant and only colonizer and significantly fewer bacteria with pathogenic potential were found with lower incidence of bacterial vaginosis in the estrogen replacement group
Raz et al. [74]	1993	93 postmenopausal women with history of recurrent urinary tract infection (UTI) randomly divided into two groups (placebo vs. estrogen)	Intravaginal estriol cream	Topically vagina	Significant decrease in incidence of UTI and vaginal lactobacilli reappeared with decrease of the rate of vaginal enterobacteriaceae colonization in estrogen treatment group
Yoshikata et al. [88]	2022	35 premenopausal and 35 postmenopausal healthy women	Lactobacillus-containing feminine soap and cream or Lactobacillus-containing feminine gel in addition to soap and cream	Topically vagina	Improvement of genitourinary symptoms and creation of a better balance of *Lactobacillus* and pathogenic flora population, especially in the postmenopausal women with feminine hygiene products and gel containing *Lactobacillus*
Jaisamrarn et al. [89]	2013	87 postmenopausal women with vaginal atrophy symptoms randomly divided into two groups (placebo vs. experimental)	Vaginal tablet (estriol 0.03 mg in combination with viable Lactobacillus acidophilus KS400)	Vagina	Significant improvement in vaginal flora, maintaining the improved maturation of the vaginal epithelium and preventing relapse of symptomatic vaginal atrophy

## Data Availability

No new data were created or analyzed in this study.

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
