# Peer review of "Menopausal Changes in the Microbiome—A Review Focused on the Genitourinary Microbiome"

_diagnostics, 2023, doi:10.3390/diagnostics13061193_

Round 1
Reviewer 1 Report
Dear Authors,
With very much interest, I have read your review, Menopause: Changes of the Genitourinary Microbiome. The article is well-written and easy to follow. I have only minor comments that might help you to improve the paper. The review is comprehensive in describing the microbiome in postmenopause, and it is stated that it is associated with many urinary tract diseases. However, the pathogenesis of this association might be addressed more. The relation between changes in the microbiome and recurrent cystitis is described profoundly; however, the pathophysiological association with interstitial cystitis remains unclear. Moreover, about 20% of the review represents the gut microbiome, even though the article's title does not refer to it.
Author Response
- Thank you for your comments.
This review was aimed at summarizing the changes in the urinary microbiome after menopause. However, because no study on the characteristics of the urinary microbiome, focused only on postmenopausal women with IC/BPS, has been conducted, it is difficult to fully explain the effect of the postmenopausal urinary microbiome on the pathophysiology of IC/BPS. However, considering the fact that the estrogen level is closely related to the symptoms of IC/BPS and based on the findings in some studies that the urinary microbiome differs significantly in IC/BPS patients, the postmenopausal situation, wherein both the estrogen level and urinary microbiome changes, is expected to affect the pathophysiology of IC/BPS. We have added relevant information to the manuscript text.
Moreover, accepting your opinion, we have revised the title to make it more comprehensive.
Reviewer 2 Report
The submitted manuscript may be interesting and useful, but does not really contribute to the literature as it stands. I propose a careful, in-depth revision. 1) The title suggests addressing postmenopausal "changes" in the urogenital microbiome. Changes to what? Please summarize the human urogenital/ vaginal microbiome at different life stages (prepubertal, reproductive years, pregnancy, postmenopause) in the form of a table or a figure. What are the main differences? For most current information I suggest reading and discussing e.g. a) Lehtoranta eta l. Healthy Vaginal Microbiota and Influence of Probiotics Across the Female Life Span. Front Microbiol. 2022 PMID: 35464937 b) Günther et al. Vaginal Microbiome in Reproductive Medicine. Diagnostics (Basel). 2022, PMID: 36010298 c) Petricevic et al. Characterisation of the oral, vaginal and rectal Lactobacillus flora in healthy pregnant and postmenopausal women. Eur J Obstet Gynecol Reprod Biol. 2012, PMID: 22088236.
2) A major shortcoming of the review is the complete omission of the role of the vaginal microbiome in oncological diseases typical of the postmenopause, especially endometrial cancer, despite rapidly accumulating evidence, e.g. a) Walsh et al. Postmenopause as a key factor in endometrial cancer microbiome (ECbiome) composition. Sci Rep. 2019, PMID: 31844128. b) Gressel et al. Characterization of the endometrial, cervicovaginal, and anorectal microbiota in postmenopausal women with endometrioid and serous endometrial cancer. Plus One. 2021, PMID: 34739493 c) Walther-Antnio et al. Possible contribution of the uterine microbiome to the development of endometrial cancer. Genome Med. 2016, PMID: 27884207. d) Wang et al. Endometrial microbiota from endometrial carcinoma and paired pericancer tissues in postmenopausal women: differences and clinical relevance. Menopause. 2022, PMID: 36150116.
3) Several relevant studies on the postmenopausal vaginal microbiome were not included in the review, why? For example, a) Petricevic et al. Differences in vaginal lactobacilli in postmenopausal women and influence of rectal lactobacilli. Climacteric. 2013, PMID: 23113473 b) Petricevic et al. Randomized, double-blind, placebo-controlled study of oral lactobacilli to improve vaginal flora in postmenopausal women. Eur J Obstet Gynecol Reprod Biol. 2008, PMID: 18701205.
4) Interventions that can help restore the postmenopausal vaginal microbiome have been partially addressed, e.g. in lines 166-178. However, I propose a separate section to summarize and discuss the possibilities, rationale, and outcomes of therapeutic and prophylactic interventions affecting the postmenopausal urogenital microbiome.
Author Response
The submitted manuscript may be interesting and useful, but does not really contribute to the literature as it stands. I propose a careful, in-depth revision. 1) The title suggests addressing postmenopausal "changes" in the urogenital microbiome. Changes to what? Please summarize the human urogenital/ vaginal microbiome at different life stages (prepubertal, reproductive years, pregnancy, postmenopause) in the form of a table or a figure. What are the main differences? For most current information I suggest reading and discussing e.g. a) Lehtoranta eta l. Healthy Vaginal Microbiota and Influence of Probiotics Across the Female Life Span. Front Microbiol. 2022 PMID: 35464937 b) Günther et al. Vaginal Microbiome in Reproductive Medicine. Diagnostics (Basel). 2022, PMID: 36010298 c) Petricevic et al. Characterisation of the oral, vaginal and rectal Lactobacillus flora in healthy pregnant and postmenopausal women. Eur J Obstet Gynecol Reprod Biol. 2012, PMID: 22088236.
-> Thank you for your comment. As suggested by you, we have included a table summarizing the changes in the vaginal microbiome at different stages of life.
2) A major shortcoming of the review is the complete omission of the role of the vaginal microbiome in oncological diseases typical of the postmenopause, especially endometrial cancer, despite rapidly accumulating evidence, e.g. a) Walsh et al. Postmenopause as a key factor in endometrial cancer microbiome (ECbiome) composition. Sci Rep. 2019, PMID: 31844128. b) Gressel et al. Characterization of the endometrial, cervicovaginal, and anorectal microbiota in postmenopausal women with endometrioid and serous endometrial cancer. Plus One. 2021, PMID: 34739493 c) Walther-Antnio et al. Possible contribution of the uterine microbiome to the development of endometrial cancer. Genome Med. 2016, PMID: 27884207. d) Wang et al. Endometrial microbiota from endometrial carcinoma and paired pericancer tissues in postmenopausal women: differences and clinical relevance. Menopause. 2022, PMID: 36150116.
-> Thank you for your comment. We have added the clinical relevance of the postmenopausal microbiome in endometrial cancer, referring to the articles suggested by you.
3) Several relevant studies on the postmenopausal vaginal microbiome were not included in the review, why? For example, a) Petricevic et al. Differences in vaginal lactobacilli in postmenopausal women and influence of rectal lactobacilli. Climacteric. 2013, PMID: 23113473 b) Petricevic et al. Randomized, double-blind, placebo-controlled study of oral lactobacilli to improve vaginal flora in postmenopausal women. Eur J Obstet Gynecol Reprod Biol. 2008, PMID: 18701205.
-> Thank you for suggesting some really valuable references. We have added a summary of these studies in the manuscript.
4) Interventions that can help restore the postmenopausal vaginal microbiome have been partially addressed, e.g. in lines 166-178. However, I propose a separate section to summarize and discuss the possibilities, rationale, and outcomes of therapeutic and prophylactic interventions affecting the postmenopausal urogenital microbiome.
-> Thank you for your comment. This review focuses on the “change” in the postmenopausal genitourinary microbiome, and more on the urinary microbiome, for which research and acquisition of evidence has been relatively slower. Therefore, an independent section on therapeutic and prophylactic interventions for postmenopausal vaginal and urinary microbiome was not included because we believe that maintaining the current section as such would provide more clarity to readers. Instead, for the benefit of readers, we have included a separate table in which we have summarized therapeutic interventions related to the vaginal and urinary microbiome.
Round 2
Reviewer 2 Report
The manuscript improved a lot.